# Review of Literature and Recommended Procedures for Management of Unusual Cases of False Positive Troponin Tests

**DOI:** 10.3390/ijms26031045

**Published:** 2025-01-26

**Authors:** Michela Salvatici, Carmen Sommese, Massimiliano M. Corsi Romanelli, Lorenzo Drago

**Affiliations:** 1UOC Laboratory of Clinical Medicine with Specialized Areas, IRCCS MultiMedica Hospital, 20099 Milan, Italy; 2Medical Direction, IRCCS MultiMedica, 20099 Milan, Italy; carmen.sommese@mutimedica.it; 3Department of Biomedical Sciences for Health, University of Milan, 20122 Milan, Italy; mmcorsi@unimi.it; 4Department of Clinical and Experimental Pathology, IRCCS Istituto Auxologico Italiano, 20095 Milan, Italy; 5Clinical Microbiology and Microbiome Laboratory, Department of Biomedical Sciences for Health, University of Milan, 20122 Milan, Italy

**Keywords:** troponin, false positive, heterophile antibodies, cardiovascular risk

## Abstract

Heterophile antibodies are immunoglobulins produced by the immune system in response to exposure to animal and bacterial antigens, blood transfusions, autoimmune disorders, hematologic malignancies, dialysis, and pregnancy. Recently, these antibodies have garnered significant attention due to their impact on the accuracy of laboratory test results. Heterophile antibodies can bind not only to specific antigens but also to those from different species, including the antibodies used in laboratory tests. This cross-reactivity with foreign proteins is the basis for their interference in immunological assays, such as those measuring cardiospecific troponins (cTn). This manuscript reviews the literature on cases of heterophile antibody interference in troponin testing and proposes an algorithm for identifying such interference when clinical discrepancies arise. Recognizing and addressing heterophile antibody interference is crucial, particularly for tests like those for troponins, which are key biomarkers in the diagnosis and management of emergency and intensive care patients. The literature emphasizes the need for accurate procedures to distinguish true cardiac damage from false positives, thereby preventing unnecessary additional tests and hospitalizations.

## 1. Introduction

Cardiac troponins have undergone one of the most rapid and innovative technological evolutions in the history of laboratory diagnostics, significantly transforming the clinical management of patients. Since their introduction into clinical practice in the 1990s, cardiac troponins have reshaped our understanding of myocardial infarction (MI) pathophysiology, revolutionized the diagnostic algorithm for acute myocardial infarction (AMI), and enhanced the monitoring of patients with acute coronary syndromes (ACSs) [1,2,3,4].

Troponins are components of the contractile apparatus in both skeletal and cardiac myocytes, playing a key role in the regulation of actin–myosin filament interaction. The cardiac troponin complex consists of three isoforms: troponin C (TnC), troponin I (TnI), and troponin T (TnT). In cardiac myocytes, about 4% to 5% of troponin is located in the cytosol, responsible for the initial surge of troponin following myocardial injury. The remainder of troponin resides in the sarcomere, where it is released slowly and continuously over several days after cell injury. Notably, troponins I and T are more specific to cardiac myocytes than TnC, making elevated levels of TnI and TnT highly specific indicators of myocardial damage [5,6,7,8].

The recognition of troponin as a pivotal player in cardiac physiology laid the foundation for its use as a biomarker. In the late 20th century, researchers discovered that cardiac-specific isoforms of TnI and TnT could be detected in the blood following myocardial injury. This breakthrough provided a more sensitive and specific marker for myocardial infarction compared to previous biomarkers like creatine kinase-MB (CK-MB) [3,8]. As a result, troponin became integral to the clinical management of MI, leading to its inclusion in diagnostic guidelines. Although troponin measurement has been recommended for AMI diagnosis since 2000, it is only in the last decade that significant improvements have been made in the analytical performance of immunoassay methods for measuring TnI and TnT [1,2,3].

Cardiac troponin assays, originally developed for diagnosing acute MI, have since become the standard biomarkers referenced in the Universal Definition of MI [1,2,3,9,10]. Over time, troponin assays have become increasingly sensitive, with newer assays capable of detecting much lower concentrations of troponin. This has significantly reduced the time window required to diagnose MI, from 6–12 h to just 1–3 h, enabling quicker and more accurate decision making in emergency settings [1,3,11,12,13,14,15,16].

The journey of cardiac troponins—from basic molecular understanding to widespread clinical application—continues to evolve. Recent advances, as reported by Clerico et al. [17], include the development of high-sensitivity troponin assays for point-of-care testing (POCT). These assays offer near-instant results, which is particularly valuable in acute settings where time is critical for patient outcomes. Early detection of elevated troponin levels enables timely initiation of treatments like reperfusion therapy in MI patients, improving clinical outcomes.

## 2. Main Laboratory/Clinical Issues Affecting Troponin Diagnosis

Despite the use of troponins being crucial for the elaboration of specific diagnosis and therapeutic strategies in current medical practice, their employment is not without challenges. Clinicians must carefully consider the clinical context to avoid overdiagnosis and unnecessary treatments. Although elevated troponin levels are most commonly associated with MI, they can also be indicative of other conditions, such as myocarditis, heart failure, pulmonary embolism, Takotsubo cardiomyopathy, cardiomyopathies, cardiotoxicity by chemotherapy treatment arrhythmias, valvular heart disease, and cardiac contusions, among others. Furthermore, ongoing research is exploring the role of troponin in various clinical scenarios and in the context of non-cardiac conditions, including renal failure, sepsis, anemia, hypotension, hypoxia, and non-cardiac surgery [18,19,20,21,22,23,24,25]. Although troponin detection in these scenarios is sometimes referred to as a “false positive”, this is incorrect, as elevated troponin levels in plasma always reflect true cardiac myocyte injury. In the general population, high cardiac troponin concentration within the normal range is associated with increased CVD risk. Indeed, literature data show the ability of hs-cTn to identify patients at low or high cardiovascular risk even with hs-cTn values below the normal cut-off [26]. More rarely, however, false troponin results may occur. Analytical interferences can arise from various factors, including fibrin clots, biotin, hemolysis, lipemia, rheumatoid factors, high alkaline phosphatase, bilirubin, heterophile antibodies, and other autoantibodies [27,28,29,30,31,32,33,34,35,36,37,38,39,40,41,42,43,44,45,46,47,48] (Table 1). Although most clinical laboratories have processes in place to check for pre-analytical errors or interferences and report significant findings, detecting analytical errors remains challenging. Antibody interference in immunoassays can affect all analytes and assays, regardless of the manufacturer. The key to identifying this interference lies in clinical observation—particularly when test results do not align with the clinical presentation, or when troponin concentrations are consistently measured as static, with no dynamic changes.

This paper reviews the literature on clinical cases of interference from heterophilic antibodies in troponin testing and proposes a systematic approach for evaluating and managing potential laboratory interference. The goal is to prevent unnecessary invasive procedures and treatments for patients if this rare, yet possible, occurrence arises.

## 3. Clinical Cases in Literature

In this literature review, we aimed to examine the interference of heterophilic antibodies and human antimurine antibodies in cardiospecific troponin testing. To identify relevant studies, we conducted a comprehensive search in the PubMed database using the query “Heterophile antibodies” combined with “troponin” and “false”. The search was limited to studies published between 2014 and 2024 to ensure the inclusion of the most recent findings. The initial search retrieved 20 records. After applying a filter for case reports, 13 studies remained. We then excluded one study published in Spanish, leaving twelve articles for eligibility assessment. Of these, one study was further excluded because it focused on false troponin results unrelated to our research topic. Ultimately, 11 studies met all inclusion criteria and were selected for this review [49,50,51,52,53,54,55,56,57,58,59] (Figure 1).

Lakusic et al. [49] published a 53-year-old female case with a history of hypertension, mild dyslipidemia, chronic stress, and smoking and referred to the county hospital as suspected ACS. The patient was hospitalized due to elevated troponin I (1359 ng/L) and underwent several invasive cardiological investigations and biomarker monitoring. Troponin I levels remained at a “plateau” during multiple measurements. However, given her clinical condition and the absence of ischemic findings, the possibility of false positive troponin due to heterophile antibodies was suspected. This was later confirmed when troponin T levels were measured in the same sample and found to be within the normal range.

Other researchers [50,51,52] have described three clinical cases regarding the influence of heterophile antibodies on troponin outcomes determined by the analyzer Access AccuTnI+3^TM^ (Beckmann Coulter Inc. Brea, CA, USA). In particular, Santos et al. [50] reported a case of a 57-year-old female patient with stable TnI elevation following suspected myocarditis. The patient was admitted to the hospital with retrosternal pain radiating to the left arm, lasting over 3 h. Clinical examination showed no significant findings, except for a history of flu illness two weeks before. ECG showed abnormal T waves and elevated creatine kinase and TnI (6.24 ng/mL) detected on Access AccuTnI+3^TM^. Coronary angiography showed no blockages, and myopericarditis was diagnosed and treated. Four weeks later, the patient returned with similar symptoms, and TnI levels had risen to 10.46 ng/mL. Coronary angiography was normal, but TnI further increased to 26.81 ng/mL. The suspected heterophilic antibodies as the reason for the elevated TnI were later confirmed by sending the blood sample to the AccuTnI+3^TM^ supplier’s laboratory to treat the interference with blocking proteins.

Similarly, Nguyen et al. [51] described a clinical case regarding a false positive elevation of TnI, measured on Access AccuTnI+3^TM^, in a 52-year-old male patient with a history of alcohol abuse, hypertension, and coronary artery bypass graft at age 34. The patient visited the emergency department due to chest pain, and upon admission, TnI levels were elevated and remained consistently high throughout the hospital stay, without showing any upward or downward trend. The patient’s ECG, other diagnostic tests, including echocardiography, and heart catheterization showed no signs of ischemia or heart muscle damage during the entire period of hospitalization. Due to the contradictory results, the accuracy of the initial TnI test was questioned and the patient’s serum sample was sent to a different laboratory, where the Advia Centaur XP TnI-Ultra (Siemens Healthineers, Malvern, PA, USA) assay showed a negative result for troponin.

Lastly, the study by Ayan et al. [52] describes the case of a 94-year-old man who was admitted multiple times to a coronary care unit due to persistently elevated TnI levels, despite the absence of clinical evidence of AMI. The patient had a history of coronary artery disease and a pacemaker but did not present chest pain or other cardiac symptoms during his admissions. Laboratory tests revealed consistently high TnI levels using the AccuTnI+3^TM^ assay, but subsequent tests using the Elecsys Troponin I assay (Roche, Diagnostic, Indianapolis, IN, USA) found undetectable levels of TnI, confirming that the elevated results were a false positive.

In 2020, two papers [53,54] discussed cases of false positive cardiac biomarkers due to interference in immunoassays, proposing a specific treatment of polyethylene glycol (PEG) precipitation and heterophile blocking tubes (HBTs) in order to successfully remove the interference and normalize TnI level.

Lewis et al. [53] reported a case of influence on the concentration of troponin I due to heterophile antibodies in a 77-year-old woman. The patient was admitted to an emergency department with chest pain and elevated TnI levels (up to 2527 ng/L), while TnT levels remained normal. This raised suspicion of assay interference. Through various tests, including treatment with PEG, it was determined that heterophilic antimouse IgG1 antibodies caused the false positive TnI elevation. Treatment with purified mouse IgG1 monoclonal antibodies finally removed the interference, normalizing the TnI levels. Differently, Hu et al. [54] described a case of a 67-year-old woman with various illnesses (e.g., autoimmune disease, glomerulonephritis) showing elevated TnI (11 ng/mL) and CK-MB (262 ng/mL) when tested with the AQT90 FLEX analyzer (Radiometer Medical Aps, København, Denmark). However, further results from the central laboratory (using the Beckman Coulter, Inc. Brea, CA, USA, DxI800 analyzer) demonstrated normal levels and the elevated biomarkers were attributed to the interference of heterophilic antibodies. Treatment with heterophilic blocking tubes drastically reduced the false positive levels of TnI and CK-MB, mitigating the interference observed in the previous immunoassay.

More recently, papers [55,56] published in 2024 described case reports about the influence of heterophile antibodies on high- and ultra-sensitive troponins assays. In the first, Millhouse et al. [55] presented three cases of elevated cardiac troponin I due to heterophile antibodies in male patients aged 50 to 70 years. They highlighted how this rare scenario can complicate the evaluation of chest pain and suspected acute coronary syndrome (ACS).

Our recent publication [54] describes an interesting case of false positive troponin value due to heterophile antibodies in a 37-year-old patient with chest pain. Stratus^®^ CS Siemens Healthineers Diagnostics permanently revealed a marked increase in TnI measurements, not confirmed by an alternative TnI assay (hs-TnI Atellica CI, Siemens Healthineers). The patient, based on the elevated TnI, underwent normal cardiological investigations without showing any cardiac damage or distress. Since the symptoms continued to be present despite the negative hs-TnI result, the patient underwent further cardiac MRI, which again reported no signs of acute myocardial edema or late gadolinium enhancement. Finally, the patient clinically improved without recurrence of chest pain and without further ECG abnormalities or abnormal echocardiographic results. Based on this clinical picture, since the elevated TnI levels did not align with the patient’s clinical condition, the presence of an interfering substance was suspected, and further investigations were planned. Three aliquots were prepared from the sample that had tested positive on Stratus CS: (a) one had been sent to an external laboratory using different hs-TnI instrumentation (Alinity ci, Abbott Laboratories, Chicago, IL, USA), which gave a negative result; (b) a second aliquot had been retested on Stratus CS after treatment with heterophilic antibody sequestration tubes (HBTs), showing a 90% reduction in TnI levels; (c) the third sample had been tested on Atellica VTLi (Siemens Healthineers, Malvern, PA, USA) for a POCT hs-TnI measurement, yielding a negative result (6.8 ng/L, reference interval < 27.1 ng/L). The presence of heterophilic antibodies was then confirmed, and the patient was discharged on the third day as a case of a false positive.

Considering the results described above, almost all the testing systems used for detecting TnT and TnI are influenced by heterophilic antibodies. Therefore, it is not yet possible to identify the most reliable testing system that is minimally affected by heterophilic antibodies. Additionally, there is currently no method that is 100% reliable in neutralizing the effects of these antibodies.

This is evident in the paper of Cheng F. et al. [57] but mainly in that of Baroni S. et al. [58]. The article of Cheng et al. [57] discusses the case of a 14-year-old girl with a diagnosis of depression who presented during a routine blood test an elevated high-sensitivity cardiac troponin T (hs-cTnT) level. This parameter was not supported by other cardiac biomarkers such as (CK) or pro-B-type natriuretic peptide (pro-BNP), which were within normal limits. Additionally, physical examination, echocardiography, and ECG revealed no abnormalities and no symptoms suggesting cardiac injury. Based on these discrepancies, the sample was reanalyzed but again showed a significant and persistence elevation in hs-cTnT levels. Therefore, it was sent to three external laboratories for further cardiac troponin testing. The first laboratory reported an elevated hs-TnT value of 880.5 ng/L (Cobas e 601 Roche, Diagnostic, Wixom, MI, USA), while the second one found a normal hs-TnI value of 1 ng/L using the Siemens system. The third reported an hs-TnI value below the detectable threshold when tested with the Abbott system. The case raised suspicion of analytical interference when a Roche analyzer was used for the hs-cTnT test. Following further analyses, including serial dilution, PEG precipitation, and the use of heterophile antibody blocking reagents, the tests concluded that the elevated hs-TnT was due to the interference from heterophile antibodies, which falsely cross-linked with the assay’s antibodies. PEG treatment had significantly reduced hs-TnT concentrations, further confirming the presence of this interference. However, the same study showed that other methods, such as high-speed centrifugation and Western blotting, were inconclusive in the investigation of heterophile antibody interference.

Baroni et al. [58] reported a case involving a 52-year-old man who experienced chest pain radiating to his left arm. His physical examination was normal, and the ECG showed a sinus rhythm with no significant ST segment or T wave changes. The patient had elevated blood pressure (170/90 mm Hg) and the laboratory tests, including creatinine and creatine kinase levels, were within the normal range. Cardiac troponin I, measured using the standard Siemens TnI-ultra test (with a reference limit of 0.040 µg/L), gave negative results both at admission and three hours later, with values of 0.012 µg/L and 0.008 µg/L, respectively. However, when tested with a different high-sensitivity analyzer, the TNIH Centaur XPT Siemens (normal up to 47 ng/L), elevated troponin I levels of 129 ng/L at admission and 140 ng/L three hours later were detected. Continuous monitoring revealed that, after 6 and 12 h, the TNIH Centaur XPT Siemens analyzer again showed elevated troponin I levels (132 ng/L and 128 ng/L, respectively), while the Siemens cTnI-ultra consistently gave negative results. To investigate the cause of these discrepancies, the patient’s serum was serially diluted with that containing undetectable troponin levels to assess result linearity. The serial dilutions revealed non-linear results when using the TNIH Centaur XPT kit, indicating the presence of an interfering substance in the patient’s sample. Moreover, since interferences can have unpredictable degrees of impact depending on the assay protocol used, the authors decided to evaluate how the patient’s serum behaved across different troponin kits in order to better understand which assays were impacted by the interfering molecule. The serum sample, which had a TnI-ultra level of 0.012 μg/L and a TNIH Centaur level of 129 ng/L, was reanalyzed using the following methods: hsTnT (Elecsys Roche, Diagnostic, Indianapolis, IN, USA), hs Pathfast cTnI (Mitsubishi Chemical Medience Corporation, Tokyo, Japan), Vidas hsTnI (BioMérieux, Marcy-l’Étoile, France), Singulex Clarity cTnI (Singulex, Alameda, CA, USA), and TNIH Dimension Vista (Siemens Healthineers, Malvern, PA, USA). Troponin levels were below the cut-off with the hsTnT, Pathfast, and Singulex kits, while high concentrations were still detected with the Vidas and TNIH Dimension kits. The most significant aspect of this article is that the interference was observed with both TNIH Siemens (Centaur XPT and Dimension Vista) using capture and detection antibodies specific to the same regions and the Vidas hsTnI kit that uses antibodies targeted to the same epitopes of the TnI-ultra kit, with which no interference was observed. This means that assays from different manufacturers can be influenced variably or similarly, depending on the epitopes targeted, as well as being affected by differences in assay architectures.

Finally, the article by Franeková et al. [59] presented a case involving a patient with chest pain and elevated hs-cTnT levels, but without clear evidence of myocardial necrosis. The findings revealed that the hs-cTnT concentration increased from 120.1 ng/L to 280.4 ng/L over an 8-month period and decreased to 216.3 ng/L in the following months. These fluctuations occurred despite repeatedly negative results of TnI, hs-cTnI, myoglobin, and CK-MB parameters. The authors employed several procedures to investigate the elevated hs-cTnT levels, but the suspected false positivity of hs-cTnT was confirmed by treating the plasma with an antiheterophile blocking agent, which reduced the hs-cTnT levels from 280.4 ng/L to 16.53/16.23 ng/L. The study concluded that the false positive hs-cTnT results were caused by the presence of an extremely rare high-molecular-weight protein. Only the pre-treatment of plasma with a blocking agent provided a reliable indication of the interference. The authors emphasized the importance of collaboration among clinicians, laboratory personnel, and manufacturers to accurately interpret such discordant results.

The case reports described here highlight a rare but important issue that should always be considered when using immunometric assays: potential interference from heterophile antibodies. These cases emphasize the need for healthcare professionals to be aware of such interferences and to adopt comprehensive diagnostic approaches when encountering discordant results.

## 4. Proposed Recommendations

The clinical cases described above demonstrated how the correct management of laboratory data and collaboration and dialogue with clinicians can be crucial to provide a correct clinical assessment of the patient. Therefore, the laboratory should adopt a critical approach to the routinely used method, understanding its strengths and limitations. When appropriate, it should evaluate the possibility of comparing with alternative methodologies or conducting further investigations and additional treatments to reliably identify any interference. In light of these considerations, a specific algorithm should be applied. Figure 2 summarizes the algorithm that can be used to manage false positive troponin results.

## 5. Discussion

Heterophile antibodies are a class of endogenous antibodies that may be produced in response to various antigens, including transfused blood, vaccinations, exposure to mice and rabbits, certain diets and medications, viral infections, rheumatoid factors, autoimmune diseases, and dialysis [60,61,62,63,64]. The exact prevalence of heterophile antibodies is unknown, and their interference with troponin assays has traditionally been considered rare. However, studies have shown that false positive troponin elevations can occur in up to 3.1% of routine populations, with a portion of these cases attributed to the presence of heterophile antibodies. Some researchers suggest that the prevalence of false results due to heterophile antibodies may increase in the future, driven by the growing use of immunotherapy for various diseases and the application of antibodies in diagnostic immunoscintigraphy studies [51].

Heterophile antibodies are weak, multispecific antibodies produced against poorly defined antigens that do not specifically bind to portions of the assay antibodies. As a result, they represent a rare but possible source of false positives in common troponin assays. This interference can be transient or permanent [60], is unpredictable, and can affect both TnI and TnT testing systems, regardless of the manufacturer. While TnI and TnT are highly sensitive and specific biochemical markers for myocardial damage, false positives should be suspected when diagnostic investigations are inconclusive or discrepant—i.e., when laboratory results do not align with the clinical picture and a static (non-dynamic) kinetic pattern of troponin is observed. Understanding the causes and mechanisms of false troponin measurements, as well as methods for identifying, confirming, and managing these interferences, has significant clinical importance.

Therefore, healthcare professionals should be aware of the potential for heterophile antibody interference in troponin testing, especially in cases where the clinical presentation and laboratory results are discordant. Early identification of such interference is crucial to avoid unnecessary treatments and procedures, ensuring accurate diagnosis and patient management. Further research and the development of more robust testing methods are needed to better detect and mitigate the impact of heterophile antibodies on troponin assays, particularly as immunotherapy and antibody-based diagnostics continue to expand.

## 6. Concluding Remarks

The technological evolution of troponin has changed the cardiovascular medicine approach. From its discovery as a biomarker of MI to the development of high-sensitivity assays, troponin has become an indispensable tool in the diagnosis, management, and prognosis of heart disease. As technology continues to advance and our understanding deepens, troponin’s role in clinical practice will undoubtedly continue to evolve, offering new opportunities for improving patient care and outcomes. However, although it is a test with consolidated clinical and analytical value, as with other immunoassays, it can be affected by interferences leading to spurious cases of false positive or false negative troponin that clinicians must consider. To address potential interference and minimize the risk of false troponin results, we strongly recommend following the proposed algorithm and recommendations outlined in this review. A collaborative approach between clinicians and laboratory personnel is essential to ensure accurate diagnosis and appropriate patient management, ultimately improving clinical outcomes and preventing unnecessary interventions, especially when diagnostic investigations are inconclusive or discrepant with the laboratory results.

## Figures and Tables

**Figure 1 ijms-26-01045-f001:**
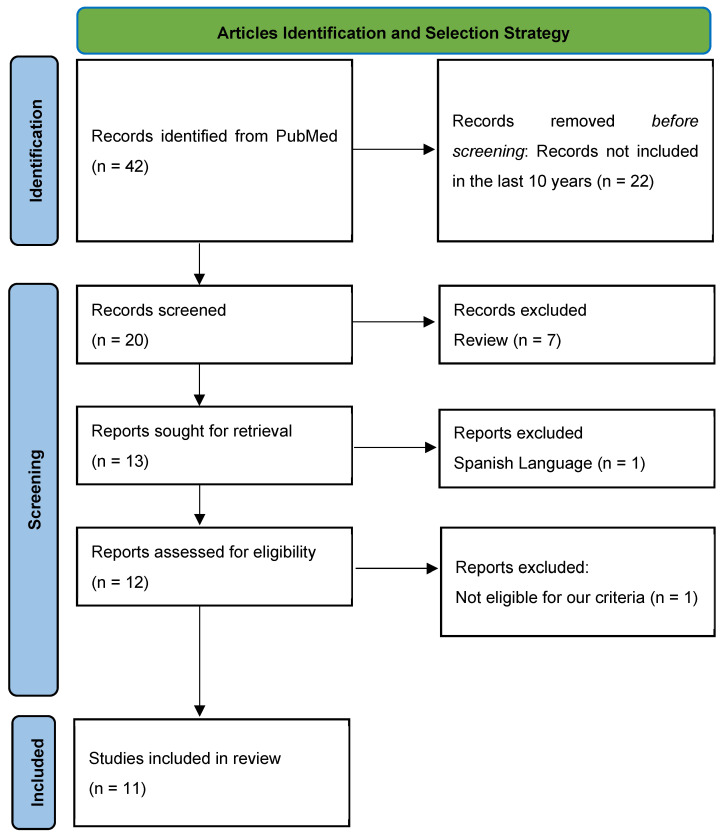
Article Identification and Selection Strategy.

**Figure 2 ijms-26-01045-f002:**
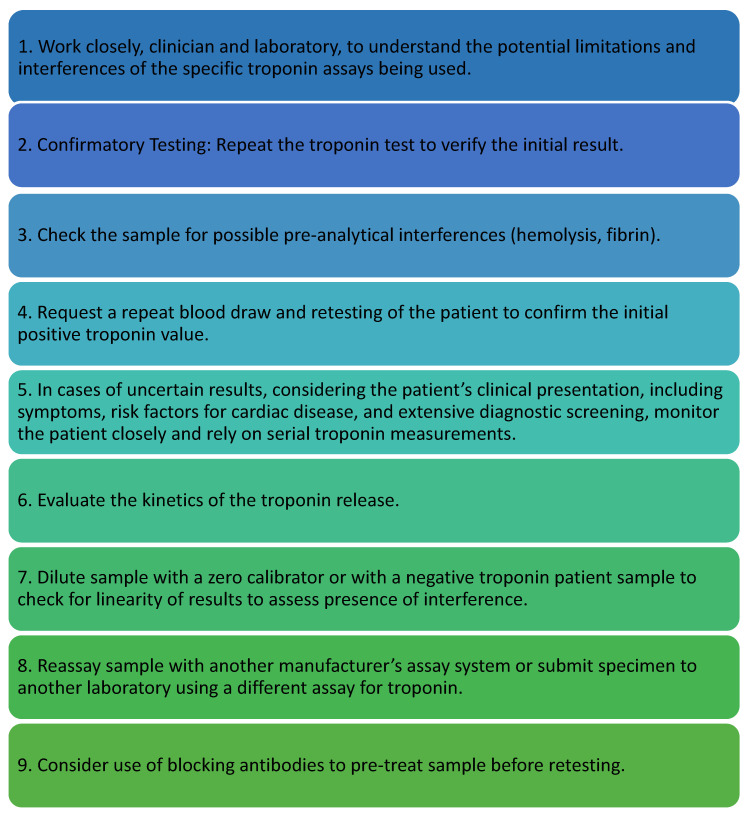
Algorithm for managing a suspected false positive troponin result.

**Table 1 ijms-26-01045-t001:** Pre-analytical and analytical causes of false negative and positive troponin values.

False Negative Troponin	False Positive Troponin
Hyperbilirubinemia or turbidity	Heterophile antibodies
Lipemia	Myopathies
Hemolysis	Hemolysis
Human antimouse antibodies	Human antimouse antibodies
Cardiac troponin autoantibodies (Macrotroponin)	Fibrin clots
Biotin (vitamin B7)	Rheumatoid factor
	Elevated alkaline phosphatase

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
