# Peer review of "Review of Literature and Recommended Procedures for Management of Unusual Cases of False Positive Troponin Tests"

_ijms, 2025, doi:10.3390/ijms26031045_

Round 1

Reviewer 1 Report

Comments and Suggestions for Authors

The paper is interesting and well developed; I suggest only few changes detailed in the attached file 

Author Response

Dear Reviewer, please see our reply on the attached file.

Thank you so much for you valuable inputs.

Best regards.

Prof. Lorenzo Drago

Reviewer 2 Report

Comments and Suggestions for Authors

Dear Authors,

I read your article „Review of Literature and Recommended Procedures for Management of Unusual Cases of False Positive Troponin Tests“ which reviews the literature on cases of heterophile antibody interference in troponin testing. It has been a popular issue throughout the years, so the authors do not bring some novelty by reviewing the thematics mentioned in the introduction. However, they carefully selected papers for this review, but this is not the first article dealing with that. They deal with cardiac troponins which represent efficient technology in the detection of severe illnesses like MI, myocarditis, and so on. The authors discussed the detection of very sensitive and specific based on troponin, more sensitive than creatine kinase. They become the standard biomarkers for the mentioned illnesses, however, it happened that troponins are specific in other illnesses like renal failure. Now is well known that false positive troponin occurs mostly concerning heterophile antibodies. The author, in detail, explains the analytical interference between heterophile antibodies and some non-specific markers like fibrin or mouse antigens

What could tell us about this kind of interference, whether it happens with some other antibodies, and what are the possible explanations of that kind of interference?

Antibody interference in immunoassays represents a great problem in the pharmaceutical industry. How hard is it to develop specific antibodies without interference with some other antigens?

Your selection of literature is excellent, you found patients with increased troponin but without MI  and authors connected them with heterophile antibodies. How hard was it to find adequate articles, and if they represent the novelty by themselves further complicated the situation?

The great question is how different authors reduce heterophilic antibodies and why they do the test in triplicates on different analyzers getting different results?

Could you give some of your own (you have a manuscript on this thematics) opinion regarding the composition of the chosen manuscripts? You choose them very thoroughly so they are more connected than heterophila antibodies.

Your proposed algorithm is excellent and I believe that could be useful to clinicians.

Conclusive remarks are a great part of the manuscript. The authors gave us their own opinions regarding the whole manuscript and selected articles. He specifically talked about the troponin and evolution of this biomarker of MI. He address the potential of interference and recommends their useful algorithm.

This is my comment regarding this very well-written, organized, and partly explained review manuscript. The questions are maybe more answers for thinking about what could be done in the future. The manuscript has great quality and interesting data.

Best regards

Author Response

(The authors gave the same response as above.)
